# Breaking Barriers: Understanding the Impact of Intellectual Impairment on Inductive Reasoning in Basketball [note 1]

**DOI:** 10.3390/sports13090318

**Published:** 2025-09-10

**Authors:** Javier Pinilla-Arbex, Javier Pérez-Tejero, Yves Vanlandewijck

**Affiliations:** 1Grupo de Investigación en Ciencias de la Actividad Física y Deporte (GICAF), Faculty of Humanities and Social Sciences, Comillas Pontifical University, 28108 Madrid, Spain; 2“Fundación Sanitas” Chair in Inclusive Sport Studies, Faculty of Physical Activity and Sport Sciences-INEF, Technical University of Madrid, 28040 Madrid, Spain; j.perez@upm.es; 3Faculty of Movement and Rehabilitation Science, KU Leuven, Tervuursevest 101, 3001 Leuven, Belgium; 4Department of Physiology, Nutrition and Biomechanics, Swedish School of Sport and Health Sciences (GIH), SE-114 86 Stockholm, Sweden

**Keywords:** eligibility, classification, para-sport, high-performance, disability

## Abstract

Access to high-performance sports is crucial for the holistic development and well-being of athletes with intellectual impairment (II). However, ensuring fair and equitable participation requires effective eligibility systems. This study investigates how basketball-specific inductive reasoning impacts athletes with II. A total of 92 elite male players with II (average age 26.3 ± 7 years) and 128 control players without II participated. This study used a tailored test to assess the ability to quickly and accurately arrange 12 photo series depicting basketball sequences. Athletes with II were hypothesized to require more time and arrange the image sequences less accurately compared to their peers. The results indicated that athletes with II took significantly more time (41.2 s ± 20.2 s) and arranged the image sequences less accurately than senior players without II (19.2 s ± 5.9 s). A discriminant function analysis classified 84.1% of players accurately, confirming that athletes with II performed at a lower level in basketball-specific activities that require inductive reasoning. These findings contribute to the development of Phase 3 of the classification model for athletes with II, which consists of 4 phases. This helps establish the eligibility system boundaries in basketball for individuals with II, promoting equitable access for athletes to high-performance sports.

## 1. Introduction

For an athlete, the pursuit of high performance represents not only a personal challenge but also a vital opportunity for growth, shaping both their career and their life. Achieving excellence in sports is not just about physical prowess—it is integral to holistic development, fostering critical benefits in physical, psychological, and social well-being [1]. Every athlete deserves the right to pursue a professional career, with access to the necessary resources, infrastructure, and opportunities that support their journey. Unfortunately, barriers remain in sports for athletes with intellectual impairment (II) [2]. Denying access or restricting opportunities due to II undermines not only their athletic potential but also their personal growth and overall health. A comprehensive sports structure is essential to ensure that athletes are not limited by their challenges but are empowered to compete, develop, and thrive in all aspects of their lives. This approach opens the door for athletes to realize their full potential, positively impacting their well-being and contributing to their development as individuals [3].

However, during the 2000 Sydney Paralympics, an international scandal emerged when athletes without II were found competing in the Games. As a result, sports for athletes with II were removed from the Paralympic program until reliable eligibility systems were developed to ensure that only athletes with significant limitations due to II were allowed to participate in these competitions [4]. Such systems are crucial to ensure fair competition in Paralympic sports [5]. However, the removal of II-sports from the Paralympics had a negative impact on the athletes with II, as participating in the Paralympic Games is the highest expression of sporting excellence at the global level [6]. To reintegrate sports for individuals with II into the Paralympic Games, since 2000, various expert groups have been investigating how II impacts performance in each sport, aiming to establish clear eligibility criteria and define the boundaries for participation [7].

In 2021, Van Biesen et al. [8] developed a conceptual model of sport-specific classification for para-athletes with II that consisted of 4 phases:Phase 1—Eligible impairment: Athletes need to present IQ ≤ 75 and adaptive behavior challenges at age onset < 22.Phase 2—Generic sport intelligence test (GSIT): Athletes must present significant limitations in cognitive abilities identified as relevant for sport performance.Phase 3—Sport-Specific Test (STT): Athletes must demonstrate significant limitations in performing fundamental sport-specific tasks, especially in tasks with high cognitive demands.Phase 4—Game observation: Athletes must present cognitive challenges in performing the sport during a real game.

Phases 1 and 2 were considered generic, while Phases 3 and 4 were sport-specific. After extensive research, table tennis, swimming, and track and field were reinstated in the London 2012 Paralympic Games [7]. However, there is still no team sport included in the program. In this context, basketball for athletes with II stands out as one of the most widely practiced sports within this population worldwide. Nevertheless, sport-specific research is still required to further develop Phases 3 and 4 of the model proposed by Van Biesen et al. [8].

In recent years, several studies have analyzed the impact of II on basketball performance. Pinilla-Arbex et al. [9] found that teams of athletes with II show lower shooting efficiency and shorter ball possessions compared to athletes without II. This suggests that athletes with II may execute actions in a more rushed or less refined manner or that a higher number of errors results in more frequent possession turnovers. This observation is consistent with the findings by Sakalidis et al. [10], who noted that while athletes without II often slow down in the final quarter, teams with II tend to increase the pace by using less of the shot clock. Such a faster tempo, driven by a less controlled game rhythm, may hinder the team’s ability to manage player fatigue and maximize scoring opportunities. In basketball, optimal pacing involves using the shot clock efficiently to ensure well-executed offensive plays while conserving energy, particularly in critical moments of the game. The differences between athletes with and without II may be linked to reduced collective actions, less effective teamwork in seeking optimal shooting opportunities, and limited execution of controlled set plays [9]. These studies are relevant in demonstrating that II directly impacts game performance and have contributed to the development of Phase 4 of the model proposed by Van Biesen et al. [8].

Despite these advances, the cognitive foundations of tactical performance have not yet been thoroughly explored. Inductive reasoning—defined as the ability to detect regularities and generate general rules from specific situations—appears to be a crucial process for interpreting complex and dynamic contexts such as team sports [11]. In basketball, this involves continuously integrating information about teammates, opponents, the ball, and contextual cues to anticipate how play will evolve. For example, inductive reasoning allows players to quickly recognize tactical patterns such as defensive switches, anticipate a pick-and-roll, or identify an open teammate after repeated ball movements. By extracting these regularities, athletes can generate predictions that guide rapid and effective decisions in real-game contexts. This skill is closely linked to broader psychological models such as dual-process theories, which distinguish between intuitive, fast responses (System 1) and deliberate, analytical reasoning (System 2) [12]. It is also connected to executive functions—including working memory, inhibitory control, and cognitive flexibility—which are essential for efficient decision-making under pressure [13]. Although not explicitly labeled as inductive reasoning, empirical studies have shown that expert athletes are able to identify patterns and regularities from game situations, allowing them to anticipate and respond effectively [14], an ability that aligns with inductive cognitive processes.

Previous research has linked inductive reasoning to tactical performance in the general population. For example, Raab & Johnson [15] emphasized that athletes with higher cognitive flexibility and reasoning abilities adapt better to dynamic game situations, generating more effective tactical solutions. However, in the case of athletes with II, the literature highlights persistent difficulties in these processes, which may compromise their capacity to extract relevant patterns from the game and anticipate opponents’ actions [8]. Thus, while the VIRTUS classification includes cognitive criteria, there is still a lack of in-depth analysis of how II specifically affects cognitive processes related to basketball.

Focusing on inductive reasoning rather than decision-making more broadly offers a novel contribution, as decision-making has already been widely studied in athletes with and without II, whereas the underlying reasoning processes that enable players to detect patterns and anticipate actions remain underexplored. By isolating this component, the present study provides unique evidence on a foundational cognitive mechanism that precedes and shapes decision-making in basketball. Furthermore, this focus carries direct practical implications: understanding the specific role of inductive reasoning may inform the development of sport-specific classification tests (Phase 3 of the classification model) [8].

In accordance with all of that, the aim of this study is to identify how II impacts inductive reasoning applied to basketball-specific tasks under testing conditions. To address this objective, two specific goals were established:To compare the performance of athletes with II in basketball-specific tests involving inductive reasoning.To explore where the performance of athletes with II stands in relation to different age groups of basketball players without II in development.

The hypotheses for these objectives are as follows: First, it is hypothesized that athletes with II will show lower performance compared to their peers without II. Second, it is expected that the performance of athletes with II will be positioned at an earlier developmental stage when compared to various age groups of basketball players without II, reflecting differences in cognitive development and basketball-related skills.

According to the World Health Organization (WHO) [16] and the International Classification of Functioning, Disability, and Health (ICF), this research also provides new tools for assessing athletes’ functionality in sport-specific contexts, promoting participation and equitable access to high-level competitions.

## 2. Materials and Methods

### 2.1. Participants

A total of 92 male basketball players with II and 128 male basketball players without II participated in this study. The group of players with II represented the entire player base from the World Basketball Championships for athletes with II (Ankara, Turkey, 2013) and 81.7% of the participants in the II-basketball competition at the Global Games (Guayaquil, Ecuador, 2015). These championships, organized by VIRTUS: World Intellectual Impairment Sport, are the highest level of basketball competition for athletes with II worldwide. Athletes from eight countries participated: Venezuela, France, Portugal, Japan, Australia, Poland, Greece, and Turkey. All players met the criteria for intellectual disability diagnosis set by the American Association on Intellectual and Developmental Disabilities (AAIDD) [17], which include significant limitations in intellectual functioning (IQ ≤ 75) and adaptive behavior deficits during the developmental period, defined as before the age of 22. However, IQ was not directly assessed in this study.

To ensure homogeneity within the group of athletes with II, eligibility had been verified by VIRTUS to participate in the international championships. This process included a review of medical and psychological documentation, ensuring that only athletes with formally diagnosed II were eligible. Although no direct IQ testing or additional neurocognitive assessments were performed by the research team, all athletes represented the international elite level of II-basketball, which implies a comparable functional level in terms of competitive participation, training demands, and exposure to high-level performance contexts.

To test the first hypothesis, a control group of senior players without II (over 18 years old) was selected, matched for experience (years playing basketball and total hours trained) and training volume (hours trained per week and months trained per year). Competitive level was controlled by selecting only athletes participating in official amateur competitions, and basketball background was recorded to confirm comparable levels across groups. For the second hypothesis, this control group was expanded to include players from various developmental stages: under-14 (U-14) years old, under-16 (U-16) years old and under-18 (U-18) years old. All players without II participated in amateur competitions in Spain or Belgium. Table 1 presents the number of participants, age, experience and training volume for each group. Experience was calculated as the product of years of training and the number of hours per year.Experience = (Years of training) × (Hours per year)

Training volume referred to the number of hours per week and the number of months of training.

**Table 1 sports-13-00318-t001:** Participants’ basketball experience and training volume.

Variable	Players with II	U-14	U-16	U-18	Senior
*n*	92	30	23	29	46
Age	26.3 (7.0)	13.2 (1.0)	15.2 (0.4)	17.7 (0.5)	23.4 (5.2)
Years playing	9.8 (6.3)	5.9 (1.9)	4.8 (2.1)	8.7 (3.5)	14.2 (5.1)
Total experience (hours)	2411 (2307)	1257 (706)	1030 (548)	1898 (1090)	2555 (1961)
Hours per week	6.3 (4.6)	6.1 (1.3)	5.4 (1.2)	5.3 (1.8)	5.4 (1.8)
Months per year	9.6 (2.2)	9.7 (0.5)	9.9 (0.3)	9.7 (0.7)	9.3 (1.8)

Standard deviations are expressed in brackets.

Power analysis was conducted using the TTestIndPower module from the statsmodels package (version 0.14.4) in Python. Assuming a two-tailed independent samples *t*-test, a significance level of α = 0.05, and a large expected effect size (Cohen’s d = 0.80), the statistical power for detecting group differences was estimated. For the full comparison between 92 participants with II and 128 without II, the analysis indicated an excellent power level of 0.99. In addition, for comparisons between the II group and the subsamples of athletes without II, the power analysis yielded values above 0.92 in all cases. These results suggest that the sample sizes used were sufficient to detect large effects with high confidence.

### 2.2. Instrument Design

A specific test was designed to evaluate the players’ ability to recognize and sequence a basketball game situation. In this test, 12 series of photographs were shown to the players, each depicting a game scenario. Each scenario was divided into 4 to 6 photographs. The sequences were administered in a physical format, printed on laminated cards, and presented to the athletes at a table. The order of photographs within each series was randomized for every trial to avoid learning effects. Their task was to rearrange them into the correct chronological order as quickly as possible (see Figure 1).

Three additional series of photographs were created: one to demonstrate the test procedure to the athletes and two trial series to ensure they fully understood the task. The photographs used in this test were taken from various game situations in Belgium’s premier basketball league, the ‘Pro League’. These situations were unanimously selected by three experienced basketball professors from KU Leuven, Belgium, each with over 15 years of expertise. The scenarios depicted in the test covered a wide range of basketball-specific concepts, including transitions, inbounds, rebounds, rules, direct and indirect screens, and situations of offensive and defensive superiority.

First, players were required to observe the images, identify the key elements in each one, connect the ideas, and determine the sequence they represent. Once this was performed, they had to arrange the images in the correct order. A correct tactical action was operationally defined as ordering the complete series in the exact chronological sequence, as validated by the expert panel. Only one correct solution was possible for each series. In addition to inductive reasoning for interpreting the sequence, working memory is essential for retaining and manipulating task-relevant information throughout the task [18]. Logical planning and sequencing were also crucial, as they enabled players to adapt to new uncertain situations, such as when the images are presented for the first time or in real-game scenarios where conditions are constantly changing [19].

During administration, the evaluator simply recorded whether the sequence arranged by the athlete matched the predefined correct order. Therefore, no subjective judgment from the test administrator was involved in the scoring process. The test was designed to progressively increase in complexity from series 1 to 12. This was achieved by introducing more challenging concepts and increasing the number of photographs in each series. Both the demonstration and the two trial series consisted of four images. In the actual test, series 1 to 6 each included four photographs, series 7 to 9 contained five photographs, and series 10 to 12 featured six photographs. The selection of photographs to represent a game situation was not based on fixed time intervals but rather on key moments in the game that effectively illustrated the tactical transition from one image to the next.

### 2.3. Procedures

The players’ task was to arrange the series of photographs into the correct sequence as quickly and accurately as possible. During the demonstration and the two trial runs, players were allowed to ask questions of the test administrator. To address the international and multilingual composition of the sample, the test was designed to be highly visual (using photographs) and required few instructions, minimizing the influence of linguistic differences. In every session, a coach or assistant was present to translate the standardized instructions when necessary, ensuring that all players clearly understood the procedure. In addition, demonstrations and trial runs were conducted to verify that the task was fully understood regardless of language or cultural background. This procedure ensured that all athletes were familiar with the demands of the test prior to its administration.

However, once the actual test began, no questions were permitted. Coaches could assist players in understanding the test’s objectives and procedures during the explanation phase, but they were not allowed to offer help or interfere during the test itself.

To present each series of photographs, players were asked to turn around while the tester arranged the images in a standardized order on a table. Once the photographs were set, the player was instructed to turn back toward the table and begin arranging them in the correct sequence. At that moment, the test administrator started the stopwatch and stopped it when the player indicated completion. After each series, the administrator recorded the time taken (to the nearest tenth of a second) and noted whether the sequence was correct. If a series was ordered incorrectly, the player’s arrangement was also documented on the score sheet. The test was conducted in a quiet room to ensure optimal concentration, and players were seated in a comfortable position to handle the photographs. No feedback regarding performance was provided during the test.

In line with ethical requirements for research with athletes with II VIRTUS informed the national teams about the study, the type of data to be collected, and its anonymous treatment. At the initial meeting, team managers and coaches were briefed on the objectives and procedures of the research. Finally, each athlete provided written informed consent prior to participating in the tests.

### 2.4. Statistical Analysis

The Kolmogorov–Smirnov test confirmed the normal distribution of the variables collected in this study (*p* > 0.05). Descriptive statistics, including means, standard deviations, and coefficients of variation, were calculated for all test variables in both groups of players, with and without II, and further stratified by age. In addition, the percentage of players who correctly ordered each photo sequence was calculated for each series in both groups.

To test the first hypothesis, a Student’s *t*-test was used to verify that there were no significant differences in training volume or experience between players with and without II. Another *t*-test was conducted to compare the results of both groups, focusing on variables such as the average time to complete the test, the time taken for each series, and the number of series correctly ordered. The effect size (ES) of these differences was calculated using Hedges’ *g* [20]. In addition, a discriminant function analysis was performed, incorporating the time required to solve each situation and the accuracy of each sequence. The structural coefficients (SCs) from the discriminant functions were used to identify the variables that best distinguished between the two groups [21]. A significance threshold for differentiation was set at SCs above |0.30| [22]. Finally, leave-one-out classification was applied to validate the discriminant models [23]; this cross-validation method classifies each case based on functions derived from all cases except the one being classified.

For the second hypothesis, the Kruskal–Wallis test was applied because some subsamples included fewer than 30 players. This test was used to explore differences across all age groups, both with and without II. Post hoc analyses were conducted using the Mann–Whitney U test to identify pairwise differences between groups. In addition, Spearman’s rank correlation analysis was performed to examine the relationship between age and overall test outcomes. All statistical analyses were carried out using PASW Statistics 24 (SPSS Inc., Chicago, IL, USA), with the significance level set at *p* < 0.05.

## 3. Results

Before testing the hypothesis, it was verified that there were no significant differences in total experience (*p* = 0.73), months of training per year (*p* = 0.58), or hours of training per week (*p* = 0.20) between players with II and senior players without II. The results showed that the average time taken by players with II to solve each series (41.2 s ± 20.2 s) was significantly longer (*p* < 0.001) than that of senior players without II (19.2 s ± 5.9 s; ES = 1.31). These differences (*p* < 0.001) were also observed across all 12 scenarios, as illustrated in Figure 2. The figure displays the mean values for each scenario along with the corresponding standard deviations, highlighting greater variability in the performance of players with II compared to those without II, as reflected in the larger standard deviations.

Regarding the number of series correctly ordered, significant differences were observed between the groups (*p* < 0.001). Players with II correctly ordered an average of 6.7 ± 3.6 series, whereas senior players correctly ordered 9.35 ± 1.5 series (ES = 0.86). Figure 3 shows the percentage of players with II and senior players who correctly solved each series. In every series, the percentage of senior players achieving the correct solution was consistently higher than that of players with II.

The discriminant function was calculated using the time taken to solve each sequence of photographs and the accuracy in ordering them. This function was statistically significant (*p* < 0.001), with a canonical correlation of 0.719. The resulting discriminant function was formulated as follows:D = t1 × 0.012 + C1 × 1.133 + T2 × 0.056 + C2 × −0.087 + T3 × 0.004 + C3 × 0.371 + T4 × −0.02 + C4 × −0.856 + T5 × −0.006 + C5 × 0.339 + T6 × 0.011 + C6 × −0.478 + T7 × 0.033 + C7 × −0.169 + T8 × −0.007 + C8 × −0.24 + T9 × 0.018 + C9 × −0.468 + C10 × −0.508 + T11 × −0.011 + C11 × −0.089 + T12 × 0.001 + C12 × 0.12 − 1.558.
where Tn = time employed to order the series “n”; C = accuracy of the order established in series “n” (0 = incorrect, 1 = correct).

When each player’s results were entered into the function, a *D value* was generated. This value indicated how closely a player’s test performance aligned with the reference values of players with II or senior players. A *D value* above –0.363 classified a participant as a player with II, whereas a value below this threshold classified them as a player without II. The function correctly classified 84.1% of players into their original groups. The structure coefficients revealed that the time required to decide on each photograph, together with the accuracy in series 10 and 11, were the key variables (|SC| > 0.30) distinguishing between players with and without II. Cross-validation using the leave-one-out method confirmed the robustness of the model, correctly classifying 78.3% of the cases.

To test the second hypothesis, the performances of players with II were compared with age-stratified samples of athletes without II using Kruskal–Wallis tests. These analyses revealed significant differences between groups in average performance time and the number of correctly ordered series (*p* < 0.001). Descriptive statistics of the test outcomes for each group, together with pairwise differences identified by Mann–Whitney U tests, are presented in Table 2.

The results from Table 2 (means and standard deviations) are depicted in Figure 4 to provide a graphical representation of the differences and to illustrate how the performance of athletes with II is positioned relative to the various subgroups of athletes without II. For each value, the groups with which there are significant differences are indicated in the graph.

Players with II took significantly more time (*p* < 0.001) to solve each sequence compared to players without II, including those in the youngest age group (U-14). In addition, athletes with II correctly ordered fewer sequences than players from the U-16 age group and older. Regarding performance progression among players without II across age groups, Spearman’s rank correlation analysis revealed a significant negative correlation between age and the time taken to solve each sequence (r_s_ = −0.326, *p* < 0.001).

## 4. Discussion

The aim of this study was to examine how II impacts inductive reasoning when applied to basketball-specific tasks under testing conditions. The first hypothesis was supported: elite players with II demonstrated significant limitations in organizing basketball game sequences as quickly and accurately as their counterparts without II, despite being matched for training volume and experience. A large effect size further underscored these differences. In line with the second hypothesis, when situating athletes with II along the developmental curve of athletes without II—from the youngest age group (U-14) to the senior category—the performance of players with II was found to be below that of U-14 players, both in terms of time and accuracy. These findings emphasize the magnitude of the impact of II and, consistent with the World Health Organization framework [16], represent progress in identifying the alteration of bodily functions and their implications for participation and health from a holistic perspective.

Inductive reasoning speed plays a crucial role in basketball performance, as it enables players to quickly identify patterns and make rapid decisions in dynamic and unpredictable game situations. According to Furley and Memmert [24], athletes with faster reasoning abilities are more capable of processing complex game information and anticipating opponents’ movements, which directly influences decision-making under pressure. In basketball, where split-second choices can determine the success of a play, a player’s ability to reason inductively allows them to adapt to evolving scenarios, such as selecting the optimal pass or defensive maneuver. Although Hohmann et al. [25] did not explicitly mention inductive reasoning in their study, they found that experts detect important action-directing cues in the environment faster and more accurately than novices.

From a sport neuropsychology perspective, these findings can be explained by integrating models of information processing and decision-making. According to Kahneman [26], fast and intuitive processes (System 1) play a central role in situations of high temporal demand, such as basketball, whereas deliberate reasoning (System 2) is often too slow. Inductive reasoning acts as a bridge between these two systems, since it allows players to derive general rules from previous experiences and to apply them flexibly in novel situations. Players with II may struggle to efficiently coordinate these systems and to engage in inductive reasoning, which could explain their slower and less accurate tactical responses. Similarly, dual-process models in decision-making in sports highlight that expertise depends on the capacity to shift flexibly between intuitive and deliberate reasoning depending on game demands—a shift in which inductive reasoning is fundamental. The impairments observed in athletes with II suggest difficulties in this flexible shift, and thus in the inductive processes required to adapt in unpredictable contexts [27].

According to the results obtained, the longer time required by athletes with II to sequence the images could negatively affect their ability to interpret the game quickly in real situations. These findings are consistent with the experimental evidence reported by Pinilla et al. [28] in decision-making field tests and in their computer-based decision-making test [29], where athletes with II also needed more time to decide. In line with this, previous research suggests that athletes with II may experience impairments in key cognitive skills involved in tactical behavior, such as inductive reasoning [24], attention allocation [25], working memory [30], and processing speed [31]. These processes are central constructs in sport neuropsychology, as they determine how efficiently athletes can perceive, interpret, and respond to game stimuli [32].

Such limitations may help explain why players with II often present different game statistics compared to athletes without II. Difficulties in rapidly recognizing game situations can lead to rushed or less elaborated tactical decisions and, consequently, a higher number of errors, such as lower shooting success [9]. Similarly, Sakalidis et al. [10] found a higher game pace among athletes with II compared to competitions involving players without II. Taken together, our findings suggest that although these players may require more time to analyze the game, when confronted with defenders or high-paced situations they are often forced to make decisions without fully understanding the context, which may result in a greater number of mistakes.

Additionally, this study examined where the performance of athletes with II falls along the developmental curve presented by athletes without II, situating the results from the athletes with II below the level of the U-14 sample. This fact underscores the significant impact that II has on inductive reasoning in specific basketball tasks for athletes competing at the highest levels of international competition. The study’s findings on the correlation between age group and test scores in players without II align with recent research, which continues to show that decision-making capacity improves with age [29], and that experts consistently outperform novices in resolving game situations more quickly [33]. Thus, the cognitive and perceptual skills analyzed in this study are essential for basketball performance, further validating the test employed.

According to the test design, it is important to highlight the distribution of the outcomes. As shown in Figure 3, in all image series there were more senior athletes without II who responded correctly to the sequences. Both curves demonstrated a certain degree of parallelism, indicating that all image series were effective in differentiating between players with and without II. The most challenging sequences for athletes with II were also the most difficult for athletes without II, which reinforces the validity of the instrument. According to Kane [34], test validity is not limited to the scores themselves but also to the interpretation of those scores in relation to the theoretical construct being measured. Kane further argues that validity must encompass both measurement and decision-making, emphasizing the importance of meaningful differentiation between groups, as observed in this study.

However, between sequences 7–8 and 9, where the number of images increased from 4 (in sequences 1–6) to 5, the difficulty did not rise progressively. This suggests a need to reconsider the order of image presentation. According to Hubley and Zumbo [35], progressive increases in task difficulty are recommended to preserve the integrity of cognitive assessments. Similarly, sequences 10–12, which contained 6 images, showed a consistent increase in difficulty for athletes with II; however, in sequence 11 athletes without II did not exhibit the same progression of difficulty observed in athletes with II.

Regarding the data distributed by age categories, Table 2 shows that U-16 athletes obtained the highest number of correct responses, although this difference was not significant when compared to the U-18 and Senior groups. This suggests that at this stage U-16 athletes may reach similar levels of accuracy in ordering the image sequences but require more time to complete the task. This interpretation is consistent with Borsboom et al. [36], who argue that variations in test outcomes are closely linked to the underlying construct being measured, which in this case reflects the developmental progression of inductive reasoning in younger athletes. These developmental differences emphasize the need to adjust test parameters to account for age-related variations in inductive reasoning ability.

In addition to classification tools development, practical applications can be derived from these findings. Coaches and sports technicians working in inclusive or adapted environments should design training tasks that incorporate cognitive stimulation strategies, such as video-based decision-making drills, small-sided games with variable constraints, or dual-task training combining motor and cognitive demands. These approaches may help athletes with II to improve their processing speed, attentional control, and inductive reasoning in ecological contexts. Moreover, the integration of neurocognitive training into basketball practice aligns with current recommendations in applied sport psychology, where cognitive skills are trained alongside technical-tactical skills to foster more adaptive decision-making [37].

As limitations of the study, it is important to highlight that the sample consists exclusively of male players, as the female population did not participate in the international basketball competitions. Additionally, the lack of detailed information regarding the type of II, IQ, and the severity of the participants represents another limitation. Access to this data would offer a more comprehensive understanding of the relationship between general intellectual functioning and the development of specific cognitive processes relevant to basketball. Moreover, the relatively small sample size may limit the generalizability of the results, and unmeasured factors such as motivation, emotional regulation, or knowledge of the rules could also have influenced the outcomes. Future research should address these aspects to strengthen the validity and applicability of the findings.

## 5. Conclusions

The results of this study provide crucial insights for developing evidence-based eligibility systems in basketball for athletes with II. Following the recommendations of the classification model presented by Van Biesen et al. [8], these systems must assess how impairment impacts key sports activities and establish a minimum level of impairment that significantly hinders performance. This study confirmed the adverse effects of II on the ability to develop specific basketball skills that rely on inductive reasoning and developed a discriminant function to classify athletes based on their test outcomes. These findings are particularly relevant to Phase 3 of the sport-specific classification model for para-athletes with II [8], as they demonstrate that cognitive processes such as inductive reasoning can serve as valid indicators for fair and accurate classification in basketball.

In practical terms, we recommend that classification panels integrate the discriminant function within Phase 3 and incorporate basketball-specific cognitive tasks to document limitations. Additionally, coaches and program designers could employ brief basketball-specific cognitive assessments to guide training and eligibility. Future research should include longitudinal studies, cognitive interventions, and analyses in other team sports to validate and extend these findings. This is essential for fostering the holistic development of high-level athletes with II while also enhancing their well-being and participation in high-performance sports.

## Figures and Tables

**Figure 1 sports-13-00318-f001:**
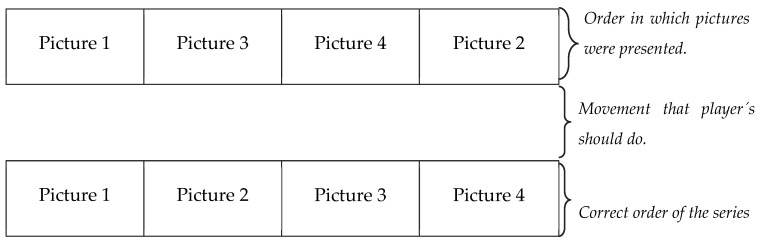
Diagram of how pictures were presented.

**Figure 2 sports-13-00318-f002:**
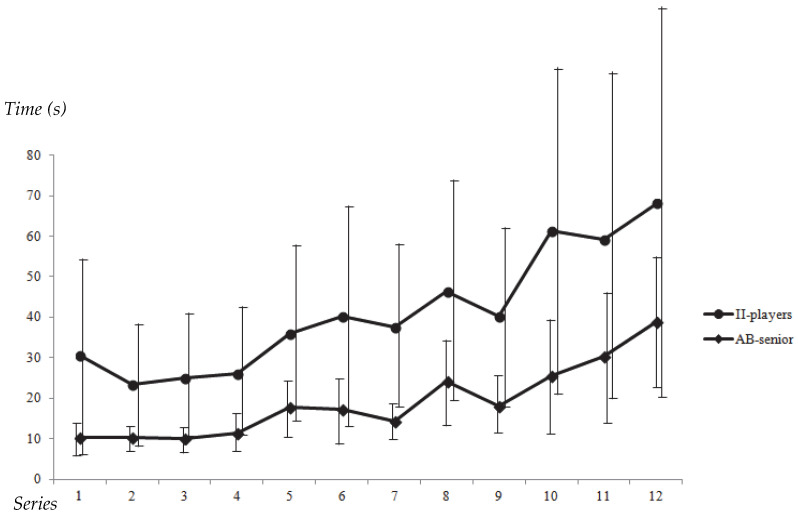
Time employed by players with II (II-players) and Senior players (AB-senior) to solve each sequence.

**Figure 3 sports-13-00318-f003:**
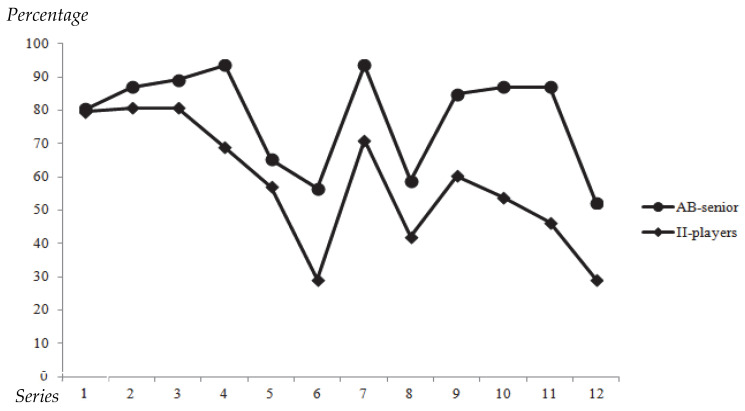
Percentage of athletes with II (II-players) and able-bodied athletes (AB-senior) that successfully ordered each series.

**Figure 4 sports-13-00318-f004:**
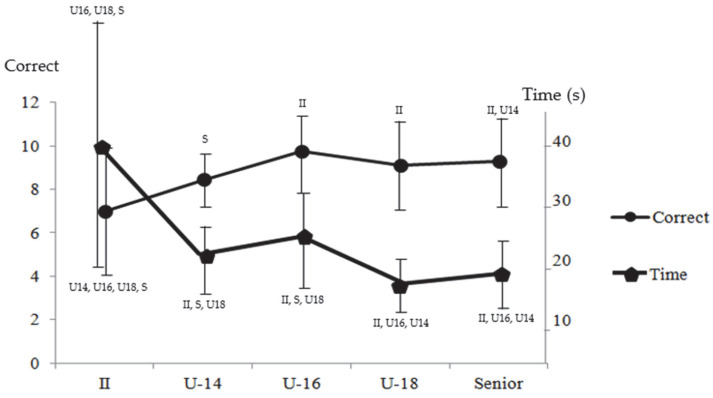
Mean time and number of correct series ordered by players with II and the different subsamples of players without II grouped by age.

**Table 2 sports-13-00318-t002:** Test scores overall comparison per competition level.

Variable	II	U-14	U-16	U-18	Senior	Significance	Effect Size
Mean Time (SD)	41.2 (20.2)	22.7 (5.5)	25.7 (7.4)	18.9 (6.7)	19.2 (5.6)	II > Senior, U18 > U16, U14	1.29/1.24/0.88/1.04
Correct (SD)	6.99 (3.4)	8.43 (1.7)	9.74 (1.9)	9.1 (1.5)	9.4 (1.5)	II < U16, U18 and Senior. U14 < Senior	0.87/0.69/0.82/0.61

SD = Standard Deviations.

## Data Availability

The data that support the findings of this study are available from the corresponding author, J.P.-A., upon reasonable request. The data are not publicly available due to privacy and ethical restrictions.

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
