# Peer review of "Breaking Barriers: Understanding the Impact of Intellectual Impairment on Inductive Reasoning in Basketball"

_sports, 2025, doi:10.3390/sports13090318_

Round 1

Reviewer 1 Report

Comments and Suggestions for Authors

The authors address a topic of great interest to researchers in the field of psychology and sport. In the introduction of their manuscript, they deal with a topic of growing interest in the field of sport psychology and inclusion: the role of cognitive abilities, specifically inductive reasoning, in tactical performance in basketball among individuals with intellectual disabilities. They provide an adequate justification for the need to understand cognitive processes in athletes with intellectual disabilities, with reference to adapted sport classification. Additionally, the relevance of the study is highlighted in the context of competition and decision-making in team sports.

However, the introduction lacks a more extensive critical review of previous studies linking inductive reasoning and tactical performance, both in the general population and in individuals with intellectual disabilities. The Virtus classification system is mentioned, but there is no in-depth discussion on how cognitive criteria relate to the specific demands of basketball. Furthermore, the theoretical framework on inductive reasoning is insufficient, as it is presented briefly and without grounding in broader psychological models (e.g., dual processing, executive functions).

Therefore, the authors are recommended to strengthen the introduction by expanding the theoretical framework on inductive reasoning and its relationship with performance in team sports. A more exhaustive review of previous works analysing the impact of intellectual disability on cognitive functions applied to gameplay should also be included.

Regarding the methodology section, the authors clearly describe the basketball player sample, distinguishing between the group with intellectual disability (ID) and the group without disability (TD), with a balanced proportion. The authors use validated tools such as the Adapted Raven’s Progressive Matrices inductive reasoning test, which provides robustness to the analysis. The experimental design (assessment phase + analysis of tactical performance through video observation) is appropriate and relevant to the objectives.

However, some recommendations are made to strengthen this section. The methodology does not mention whether a sample size calculation or a statistical power analysis was performed, which limits the evaluation of sample adequacy. Although the analysis of tactical actions is described, information is missing about the observation system, its validity and reliability, as well as the analysis criteria (what is considered a correct tactical action?). Lastly, the procedure for assigning and controlling confounding variables (such as competitive level or previous experience) is not clearly specified.

Therefore, it is recommended to include details on the reliability of the observation tools (e.g., inter-rater agreement), justify the sample size, and detail how external variables such as sports level or educational level were controlled.

As for the results, the authors clearly present the results of inductive reasoning and tactical performance, distinguishing between both groups. Appropriate inferential statistics are used (Student’s t-test and Pearson correlations), allowing the exploration of differences and associations. Significant differences are identified between groups, both in inductive reasoning and in correct tactical actions, supporting the proposed hypotheses.

However, effect sizes (such as Cohen’s d) are not reported, which limits the practical interpretation of the observed differences. No interaction or regression analyses are conducted to establish the weight of inductive reasoning as a predictor of performance. Confidence intervals and graphical representations of the results are also missing, which would help visualize the impact between groups.

The authors should include effect sizes, represent the data with comparative graphs, and, if possible, perform regression analyses or more advanced models that allow for predictive relationships.

In the discussion, the authors appropriately interpret the observed differences, highlighting the relationship between cognitive skills and tactical effectiveness in real sports contexts. They also contextualize the findings in relation to the literature on functional classification and intellectual disability.

However, the explanation of why inductive reasoning influences tactical performance could be expanded, integrating models of information processing and decision-making. The study’s limitations, such as the small sample size or the possible influence of unmeasured factors (such as motivation, understanding of rules, etc.) are not sufficiently discussed, and there are no practical proposals or applications for coaches and sports technicians in inclusive or adapted environments.

The authors should better integrate the results into the framework of sport neuropsychology and propose practical applications (e.g., cognitive stimulation programs in training). A more explicit section on the limitations of the study should be included.

Finally, the manuscript’s conclusions adequately summarize the key findings, highlighting the need to consider the cognitive profile of athletes with intellectual disabilities in evaluation and training contexts. The role of inductive reasoning as a relevant dimension for functional sport classification is emphasized.

However, the conclusions could be more operational, indicating specific courses of action in the field of adapted sport or recommendations for classification systems. Clear proposals for future research are also lacking (e.g., longitudinal studies, cognitive interventions, analysis in other team sports).

Therefore, the authors should expand the conclusions by proposing future lines of research and applications for designers of inclusive sports programs.

For all these reasons, it is recommended that the authors make a minor revision considering the provided considerations, which can be summarised as follows:

  1. Expand the theoretical framework with reasoning and decision-making models applied to sport.
  2. Justify the sample size and control for variables such as sports level.
  3. Include complementary analyses (effect sizes, graphs, regression).
  4. Discuss the limitations and propose concrete practical applications.
  5. Update and expand key references on cognition and adapted sport.

Author Response

Dear Reviewer,
Thank you for your comments and feedback. They have been very useful in improving the quality of the manuscript. Below, we provide our response to each comment and explain how we have addressed them in the document.

Comments 1: The introduction lacks a more extensive critical review of previous studies linking inductive reasoning and tactical performance, both in the general population and in individuals with intellectual disabilities. The Virtus classification system is mentioned, but there is no in-depth discussion on how cognitive criteria relate to the specific demands of basketball. Furthermore, the theoretical framework on inductive reasoning is insufficient, as it is presented briefly and without grounding in broader psychological models (e.g., dual processing, executive functions).

Therefore, the authors are recommended to strengthen the introduction by expanding the theoretical framework on inductive reasoning and its relationship with performance in team sports. A more exhaustive review of previous works analysing the impact of intellectual disability on cognitive functions applied to gameplay should also be included.

Responses 1: Following the recommendation, we have expanded the introduction to include a more comprehensive theoretical framework on inductive reasoning and its role in tactical performance in team sports. We also integrated broader psychological perspectives, such as dual-process theories and executive functions, to provide a stronger grounding. In addition, we reviewed previous studies that link inductive reasoning and tactical behavior in athletes, and we deepened the discussion on how Virtus cognitive criteria are connected to the specific demands of basketball.

Comments 2: The methodology does not mention whether a sample size calculation or a statistical power analysis was performed, which limits the evaluation of sample adequacy. Although the analysis of tactical actions is described, information is missing about the observation system, its validity and reliability, as well as the analysis criteria (what is considered a correct tactical action?). Lastly, the procedure for assigning and controlling confounding variables (such as competitive level or previous experience) is not clearly specified. Therefore, it is recommended to include details on the reliability of the observation tools (e.g., inter-rater agreement), justify the sample size, and detail how external variables such as sports level or educational level were controlled.

Responses 2: we have expanded the methodology to clarify key aspects. Specifically, we included a justification of the sample size (representativeness of the population and post-hoc power analysis). Additionally, we specified how potential confounding variables (e.g., competitive level, experience, training volume) were controlled through participant selection, statistical comparisons, and standardized testing procedures. In addition, we have clarified how data were gathered, as the test was not based in observation but in recording the order athletes provided in each series of photographs.

Comments 3: effect sizes (such as Cohen’s d) are not reported, which limits the practical interpretation of the observed differences. No interaction or regression analyses are conducted to establish the weight of inductive reasoning as a predictor of performance. Confidence intervals and graphical representations of the results are also missing, which would help visualize the impact between groups. The authors should include effect sizes, represent the data with comparative graphs, and, if possible, perform regression analyses or more advanced models that allow for predictive relationships.

Response 3: A power analysis was included to assess the adequacy of the sample sizes used, as well as the magnitude of the effect size in the comparison between independent groups. However, regarding the suggestion to perform a regression analysis, we considered that the discriminant analysis already provides a multifactorial approach, incorporating all test variables across the different images to distinguish between athletes with and without intellectual disabilities based on test performance.

One of the strengths of discriminant analysis is its ability to establish the D value as a cut-off point that best separates the centroids of the results obtained by both samples. This makes it a highly relevant statistical method for the main goal of the study: to contribute to the development of eligibility systems in basketball for athletes with intellectual disabilities (ID). As a result, a discriminant equation is generated that allows determining whether an athlete's performance in this test more closely resembles that of players with or without ID.

With regard to the potential relationship between test performance and on-court performance, we acknowledge the importance and relevance of such an analysis. However, it falls outside the scope of the present study, as performance-based game metrics were not collected or analyzed. Nevertheless, this remains a valuable avenue for future research and has been addressed in the discussion section.

Comments 4: The explanation of why inductive reasoning influences tactical performance could be expanded, integrating models of information processing and decision-making. The study’s limitations, such as the small sample size or the possible influence of unmeasured factors (such as motivation, understanding of rules, etc.) are not sufficiently discussed, and there are no practical proposals or applications for coaches and sports technicians in inclusive or adapted environments. The authors should better integrate the results into the framework of sport neuropsychology and propose practical applications (e.g., cognitive stimulation programs in training). A more explicit section on the limitations of the study should be included.

Response 4: we have included the explanation of why inductive reasoning influences tactical performance by integrating information processing and decision-making models from sport neuropsychology (e.g., Kahneman, 2011; Raab, 2012). Furthermore, we have contextualized the results within the framework of cognitive processes relevant to sport performance (e.g., executive functions; Vestberg et al., 2017). A new section on practical applications has been incorporated.  Finally, we have included a more explicit discussion of the study’s limitations, addressing sample size, lack of female participants, and unmeasured factors such as motivation or rule knowledge, which could have influenced the results.  

Comments 5: The conclusions could be more operational, indicating specific courses of action in the field of adapted sport or recommendations for classification systems. Clear proposals for future research are also lacking (e.g., longitudinal studies, cognitive interventions, analysis in other team sports). Therefore, the authors should expand the conclusions by proposing future lines of research and applications for designers of inclusive sports programs.

Response 5: we have expanded the conclusions to make them more operational and actionable. Specifically, we now include recommendations for classification systems.  Furthermore, we added proposals for future research, such as longitudinal studies, cognitive interventions, and cross-sport analyses in other team sports. These additions strengthen the applied value of the study and outline concrete avenues for advancing inclusive and evidence-based practices in adapted sport.

Reviewer 2 Report

Comments and Suggestions for Authors

The article addresses a timely and important topic concerning the cognitive functioning of athletes with intellectual impairment and their eligibility to compete in high-performance sports. The study is well-structured and grounded in a clearly defined classification model. However, several aspects in the introduction and methodology sections would benefit from further clarification and elaboration to enhance the clarity, depth, and interpretability of the work.

In the Introduction, a more detailed explanation of the theoretical relationship between inductive reasoning and basketball-specific decision-making would be useful. Although the text refers to Singh and Agashe’s definition of inductive reasoning, it does not clearly describe how this cognitive process operates in real-game contexts. A short explanation of how inductive reasoning supports behaviors such as recognizing tactical patterns or anticipating opponents’ actions would help the reader better understand its relevance. Moreover, the rationale for focusing specifically on inductive reasoning, rather than on decision-making more broadly, is not sufficiently developed. As the latter is already a well-established area of research in sports and II contexts, it would strengthen the introduction to more clearly articulate what the present study adds and why the chosen focus is unique. Additionally, the practical value of this research could be more explicitly highlighted. While it is implied that the findings may inform classification processes or athlete support strategies, a more direct discussion of how this could translate into coaching or competition policy would enhance the relevance of the work.

In the Materials and Methods section, the sample is described in detail in terms of numbers and countries of origin. However, the level of homogeneity across participants with II remains unclear, particularly in terms of variation in cognitive functioning. Since IQ was not directly assessed, it would be helpful to explain how variation in intellectual impairment was accounted for or minimized. It would also be useful to note whether additional cognitive, behavioral, or functional assessments were performed to better characterize the group.

The instrument used - the photograph-based sequencing task - is original and appears well thought out, but some details require further explanation. It is not entirely clear how the task was administered: were the sequences presented in a physical format (e.g., printed on cards) or digitally? Was the order of photos randomized for each trial or participant, and how was the “correct” sequence defined? Further, it would be important to report whether more than one correct solution was possible or whether the scoring allowed for partial correctness. If there was any subjectivity in evaluating the order, inter-rater reliability would be important to document. 

Another important consideration is the international and multilingual nature of the sample. Since participants came from eight different countries, it would be useful to explain how language and cultural variation were managed. Were the task instructions standardized and translated appropriately? Could differences in cultural familiarity with the types of actions shown in the photos affect task performance?

The statistical analysis appears appropriate and robust, and the cross-validation approach supports the classification model. Still, the manuscript could expand on the clinical or practical significance of the observed differences. That is, what does a difference in test performance imply in real-world basketball behavior or in classification decisions for athletes with II?

Btw., while inclusion criteria and recruitment procedures are outlined, the methodology lacks information regarding ethical procedures, particularly how informed consent was obtained from participants with intellectual impairment. Since this is a vulnerable population, ensuring ethical clarity is crucial.

Author Response

Dear Reviewer,
Thank you for your comments and feedback. They have been very useful in improving the quality of the manuscript. Below, we provide our response to each comment and explain how we have addressed them in the document.

Comments 1: In the Introduction, a more detailed explanation of the theoretical relationship between inductive reasoning and basketball-specific decision-making would be useful. Although the text refers to Singh and Agashe’s definition of inductive reasoning, it does not clearly describe how this cognitive process operates in real-game contexts. A short explanation of how inductive reasoning supports behaviors such as recognizing tactical patterns or anticipating opponents’ actions would help the reader better understand its relevance.

Response 1: In the revised Introduction, we have provided a more detailed explanation of the theoretical relationship between inductive reasoning and basketball-specific decision-making. In particular, we now describe how inductive reasoning operates in real-game contexts by illustrating its role in recognizing tactical patterns (e.g., defensive switches), anticipating opponents’ actions (e.g., screen-and-roll situations), and identifying open teammates after repeated ball movements. These additions clarify how this cognitive process supports the generation of rapid and effective decisions during dynamic play, complementing the theoretical models already included (dual process theories and executive functions).

Comments 2: The rationale for focusing specifically on inductive reasoning, rather than on decision-making more broadly, is not sufficiently developed. As the latter is already a well-established area of research in sports and II contexts, it would strengthen the introduction to more clearly articulate what the present study adds and why the chosen focus is unique. Additionally, the practical value of this research could be more explicitly highlighted. While it is implied that the findings may inform classification processes or athlete support strategies, a more direct discussion of how this could translate into coaching or competition policy would enhance the relevance of the work.

Response 2: In the revised Introduction, we have clarified why inductive reasoning was chosen as the focus of the study rather than decision-making in general. Specifically, we emphasize that while decision-making has been widely investigated in sports and II contexts, the underlying inductive reasoning processes remain underexplored, despite their critical role in detecting patterns and anticipating actions. By isolating this cognitive component, the present study adds a unique perspective that precedes and shapes decision-making. Additionally, we have highlighted the practical implications of our findings, explicitly linking them to sport-specific classification systems, coaching strategies, and competition policy. These changes strengthen the originality and applied relevance of the manuscript.

Comments 3: In the Materials and Methods section, the sample is described in detail in terms of numbers and countries of origin. However, the level of homogeneity across participants with II remains unclear, particularly in terms of variation in cognitive functioning. Since IQ was not directly assessed, it would be helpful to explain how variation in intellectual impairment was accounted for or minimized. It would also be useful to note whether additional cognitive, behavioral, or functional assessments were performed to better characterize the group.

Response 3: In the revised Materials and Methods section, we have clarified how variation in intellectual impairment was addressed. Specifically, we now explain that all athletes with II were pre-validated by VIRTUS through medical and psychological documentation to ensure a formal diagnosis of intellectual impairment and eligibility for international competition. Although no direct IQ testing or additional neurocognitive assessments were conducted by our research team, the fact that participants represented the entire population of elite II-basketball players at the highest competitive level provided a degree of functional homogeneity in terms of training demands and competitive exposure. We have also acknowledged in the Limitations section that the absence of direct IQ and cognitive measures restricts the analysis of intra-group variability, and we suggest that future studies should incorporate these assessments.

Comments 4: The instrument used - the photograph-based sequencing task - is original and appears well thought out, but some details require further explanation. It is not entirely clear how the task was administered: were the sequences presented in a physical format (e.g., printed on cards) or digitally? Was the order of photos randomized for each trial or participant, and how was the “correct” sequence defined? Further, it would be important to report whether more than one correct solution was possible or whether the scoring allowed for partial correctness. If there was any subjectivity in evaluating the order, inter-rater reliability would be important to document. 

Response 4: We have clarified several aspects of the photograph-based sequencing task. The sequences were administered in physical format (laminated cards), and the order of photographs was randomized for each trial. We now specify that each series had only one unique correct solution, previously validated by an expert panel, and the administrator’s role was limited to recording whether the athlete’s sequence matched this predefined order, which eliminated any risk of subjective judgment.

Comments 5:  Another important consideration is the international and multilingual nature of the sample. Since participants came from eight different countries, it would be useful to explain how language and cultural variation were managed. Were the task instructions standardized and translated appropriately? Could differences in cultural familiarity with the types of actions shown in the photos affect task performance?

Response 5:  We have expanded the Methods section to clarify how the international and multilingual composition of the sample was managed. Specifically, we now state that the test was designed to be highly visual (using photographs) and required minimal instructions, that a coach or assistant was always present to translate standardized instructions when necessary, and that demonstration plus trial runs were performed to ensure full understanding regardless of language or cultural background. These additions directly address the reviewer’s concern about standardized instructions, translation, and cultural familiarity.

Comments 6: The manuscript could expand on the clinical or practical significance of the observed differences. That is, what does a difference in test performance imply in real-world basketball behavior or in classification decisions for athletes with II?

Response 6: we have expanded the discussion to highlight the practical implications of the observed differences. Specifically, we now explain that variations in test performance reflect meaningful differences in basketball-specific skills such as the ability to interpret and read the game, anticipate opponents’ actions, and appropriately select responses in dynamic situations. These outcomes are directly relevant to the classification process, as they help to establish whether intellectual impairment significantly affects core basketball behaviors. In this way, the findings support evidence-based eligibility decisions and contribute to developing fairer and more functional classification systems for athletes with II.

Comments 7:

while inclusion criteria and recruitment procedures are outlined, the methodology lacks information regarding ethical procedures, particularly how informed consent was obtained from participants with intellectual impairment. Since this is a vulnerable population, ensuring ethical clarity is crucial.

Response 7: To address this, we have expanded the methodology section to clarify the ethical procedures followed in the study. Specifically, we now state that VIRTUS informed the national teams about the study, the nature of the data collected, and its anonymous treatment. In addition, during an initial meeting, team managers and coaches were briefed on the objectives and procedures. Finally, it is specified that each athlete provided written informed consent prior to participation.

Reviewer 3 Report

Comments and Suggestions for Authors

This study was conducted to provide data for developing a scientific classification system to enable fair participation in high-performance sports for people with cognitive impairment. The study's strengths lie in its clear purpose and the use of discriminant analysis tailored to the study's objectives, enhancing the validity of the results.

Minor revisions include the following: The Kolmogorov-Smirnov test was described in the methods section, but the results for this test were not available. The results are presented descriptively, but a table format would be helpful for easier understanding. Furthermore, the figures would benefit from noting any significant differences. More specific information on how the results can be applied in the field would be helpful. Furthermore, it would be helpful to discuss future actions to establish the results of this study as a classification system.

Author Response

Dear Reviewer,

Thank you for your comments and feedback. They have been very useful in improving the quality of the manuscript. Below, we provide our response to each comment and explain how we have addressed them in the document.

Comments 1: The Kolmogorov-Smirnov test was described in the methods section, but the results for this test were not available.

Response 1: The significance level used in the Kolmogorov-Smirnov test has been included.

Comments 2:

The results are presented descriptively, but a table format would be helpful for easier understanding.

Response 2: Data related to the athletes’ profiles and effect sizes have been incorporated into the previously existing tables. Some information has been presented descriptively rather than in table format, as these values result from specific analyses rather than from a sequence of related data.

Comments 3: The figures would benefit from noting any significant differences.

Response 3: In Figure 3, significant differences were not marked because all image sequences showed statistically significant differences. This decision is explained in the paragraph preceding the figure for greater clarity. We considered that marking each data line could clutter the figure, which is clearer when accompanied by the prior explanation. Figure 2 presents percentages; therefore, no significant differences are reported. In Figure 4, the groups with which each group shows significant differences in the variables time and number of correct responses are indicated.

Comments 4: More specific information on how the results can be applied in the field would be helpful.

Response 4: we have revised the discussion to include more concrete applications of the findings in the field. We now explain that the test can be used by coaches and support staff as a practical tool to identify athletes who may experience difficulties in interpreting game situations, and to design targeted training interventions to improve these skills. Additionally, the results can inform classifiers and sport governing bodies by providing objective evidence that supports fairer eligibility and classification procedures for athletes with intellectual impairment.

Comments 5: It would be helpful to discuss future actions to establish the results of this study as a classification system.

Response 5: We have extended the discussion and conclusion sections to highlight that the findings contribute directly to Phase 3 of the sport-specific classification model for para-athletes with II, recommend the integration of the discriminant function within classification panels, and emphasize the need for future research—including longitudinal studies and validation across international competitions coordinated by governing bodies such as VIRTUS to ensure fairness and consistency.

Round 2

Reviewer 2 Report

Comments and Suggestions for Authors

The article is much better now, thank you.